# In Vitro Antibacterial Activity of Biological-Derived Silver Nanoparticles: Preliminary Data

**DOI:** 10.3390/vetsci7010012

**Published:** 2020-01-23

**Authors:** Gabriele Meroni, Joel F. Soares Filipe, Piera A. Martino

**Affiliations:** 1Department of Veterinary Medicine, Università degli Studi di Milano, 26900 Lodi, Italy; joel.soares@unimi.it (J.F.S.F.); piera.martino@unimi.it (P.A.M.); 2Department of Biomedical Sciences for Health, Università degli Studi di Milano, 20133 Milano, Italy

**Keywords:** antibiotic resistance, green synthesis, *Pseudomonas aeruginosa*, silver nanoparticles, *Staphylococcus pseudintermedius*

## Abstract

Silver nanoparticles (AgNPs) are promising alternatives to antibiotics. The aims of this study were to produce AgNPs using two biological methods and determine their antibacterial activity against *Pseudomonas aeruginosa* and *Staphylococcus pseudintermedius*. AgNPs were biosynthesized from an infusion of *Curcuma longa* (turmeric) and the culture supernatant of *E. coli*. Characterization was achieved by ultraviolet-visible spectroscopy and by Transmission Electron Microscopy (TEM). The antibacterial properties of NPs from *C. longa* (ClAgNPs) and *E. coli* (EcAgNPs), alone and in combination with carbenicillin and ampicillin, were investigated through the Kirby-Bauer disk diffusion assay and the minimum inhibitory concentration (MIC). Dimensions of NPs ranged from 11.107 ± 2.705 nm (ClAgNPs) to 27.282 ± 2.68 nm (EcAgNPs). Kirby-Bauer and MIC assays showed great antibacterial abilities for both NPs alone and in combination with antibiotics. EcAgNPs alone showed the most powerful antibacterial activities, resulting in MIC values ranging from 0.438 ± 0.18 µM (*P. aeruginosa*) to 3.75 ± 3.65 µM (*S. pseudintermedius*) compared to those of ClAgNPs: 71.8 ± 0 µM (*P. aeruginosa*) and 143.7 ± 0 µM (*S. pseudintermedius*). The antibiofilm abilities were strain-dependent, but no statistical differences were found between the two NPs. These results suggest the antibacterial potential of AgNPs for the treatment of infectious diseases.

## 1. Introduction

The introduction of nanoscale materials in the health industry is an emerging area of nanotechnology [1,2,3]. In Europe, nanomaterials are defined as “natural, incidental or manufactured material containing particles, in an unbound state or as an aggregate or as an agglomerate and where, for 50% or more of the particles in the number size distribution, one or more external dimensions is in the size range 1 nm–100 nm” [4].

The unique characteristics of silver nanoparticles are due to their surface-to-volume ratio that considerably changes physical, chemical, and biological properties; for these reasons, these nanoparticles have been used for various purposes [5], including textiles, keyboards, wound dressings, and biomedical devices [6,7]. The size, shape, and surface coatings of nanoparticles are essential characteristics that directly determine their biocidal activities [8]. Smaller particles have a larger surface to volume ratio and, therefore, display greater toxic potential. A recent study found that the biological properties of AgNPs strictly depend on the different surface charges of their coatings, which can affect the physical interaction of AgNPs with microorganisms [9].

Many methods have been used for the synthesis of silver nanoparticles and nowadays, are categorized as: (a) chemical methods; (b) physical methods and (c) biological methods.

In the chemical methods, three main components, such as metal precursors, reducing agents, and stabilizing/capping agents, are usually employed to produce metal nanoparticles [8,10,11]. The physical approaches include direct interaction of laser or high voltage gate with raw metal, resulting in immediate nanoparticles release [10,12]. To overcome the use of hazardous chemicals and the high energy consumption required by previous described procedures, biological methods have emerged as viable alternatives. The synthesis of AgNPs using green, cost-effective, and biocompatible methods can be achieved by bacteria [13,14], fungi [15], and plant extracts [16,17,18].

Biological methods offer numerous advantages compared to both chemical and physical ones: (a) proteins and secondary metabolites are released by living organisms in the synthesis process; (b) elimination of steps to prevent particle aggregation; (c) eco-friendly and pollution-free characteristics of biological molecules are used [8].

Silver (as silver ion) has been used as an antibacterial agent for centuries until the discovery of modern antibiotics. Today, the global spread of bacteria resistant to the most common antibiotics redirects the scientific research to alternative strategies able to fight also against multi-drug resistant (MDR) bacteria. Silver nanoparticles (AgNPs) have become a target for researchers due to their antimicrobial efficacy against bacteria, fungi, and viruses. In the literature, the antibacterial properties of AgNPs against *S. aureus*, *E. coli*, *P. aeruginosa,* and *S. typhi* are described by different authors [5,18,19]. The antibacterial actions exerted by AgNPs are linked to: (a) adhesion to the cell wall, (b) penetration and damaging of cytoplasmatic organelles; (c) induction of oxidative stress via reactive oxygen species (ROS) production; (d) modulation of signal transduction pathways (e.g., stress response pathway) [20].

The dissemination of antibiotic-resistant bacteria is a global and rapidly emerging problem. In reptiles, one of the most isolated bacteria is *P. aeruginosa*, often displaying resistance to multiple veterinary prescribed antibiotics [21]. In dogs, *S. pseudintermedius* is considered one of the main causative agents of canine pyoderma able to carry Staphylococcal Cassette Chromosome *mec* (SCCmec), coding for methicillin resistance, and being able to produce several virulence factors, including biofilm [22].

The specific aims of this study are: (a) the set-up and optimization of protocols for the synthesis of silver nanoparticles using two biological methods and (b) the analysis of their antibacterial and anti-biofilm properties against *P. aeruginosa* and *S. pseudintermedius* strains isolated from animals.

## 2. Materials and Methods

### 2.1. Bacterial Strains and Culture Conditions

The bacterial strains used to study the antibacterial ability of AgNPs were randomly chosen from the bacterial collection of the Microbiology Laboratory of the Department of Veterinary Medicine of the Università degli Studi di Milano. Ten strains of *P. aeruginosa* were isolated from eyes lavages of Chameleons (*Furcifer pardalis*); sheep blood agar and Cetrimide agar (Oxoid, Italy) were used to isolate and verify bacterial isolates belonging to *Pseudomonas aeruginosa* species. Ten multi-drug-resistant (MDR) *S. pseudintermedius* strains were isolated from canine pyoderma and characterized phenotypically and genetically using, respectively, Mannitol Salt Agar (Microbiol, Italy) and the amplification of thermonuclease (*nuc*) gene as reported previously [23]. The antibiotic resistance profiles of each strain were assessed before this study with the Kirby-Bauer disk diffusion method; all the isolates were tested for amoxicillin + clavulanic acid (30 µg), amoxicillin (10 µg), ampicillin (10 µg), cephalexin (30 µg), cefovecin (30 µg), ceftiofur (30 µg), ceftriaxone (30 µg), clindamycin (10 µg), lincomycin + spectinomycin (15 µg), doxycycline (30 µg), enrofloxacin (5 µg), marbofloxacin (5 µg), pradofloxacin (5 µg), amikacin (30 µg), gentamicin (30 µg), neomycin (30 µg), tobramycin (30 µg), kanamycin (30 µg), rifampicin (30 µg), azithromycin (15 µg), erythromycin (30 µg). *P. aeruginosa* strains were also tested for carbenicillin (100 µg). All the strains were stored in 25% glycerol (Carlo Erba, Italy) at −20 °C until use. However, before use, samples were thawed at room temperature and 10 μL was plated on Tryptic Soy Agar + 1% sheep blood (Microbiol, Italy) and incubated aerobically at 37 °C for 24 h. Three or four isolated colonies were picked up and used to assess the antibacterial activity of silver nanoparticles.

### 2.2. Biosynthesis of AgNPs Using Curcuma Longa Extract

#### 2.2.1. Preparation of *C. longa* Extract

The *C. longa* powder (turmeric) was kindly provided by Dr. G. Graziani (Farmacia Graziani, Italy). The production of *C. longa* extract was assessed using a protocol found in the literature (Shameli et al., 2012) with some modifications. Briefly, after dissolving 0.1 g of *Curcuma* powder in 20 mL of double distilled sterile water (ddH_2_O) the solution was stirred for 4 h at room temperature in dark conditions and filtered (Whatman^®^ Grade 42, Ashless Filter Paper, Milan, Italy) to remove debris. This solution was used to synthesize the nanoparticles.

#### 2.2.2. Synthesis and Purification of Silver Nanoparticles

Forty millilitres of 1% AgNO_3_ (Carlo Erba, Italy) was added to the fresh *Curcuma* extract and mixed (at 200 rpm) for 24 h in the dark (to avoid photochemical reactions) and at room temperature. The reduction of Ag^+^ to Ag^0^ was monitored by checking the change in colour of the solution: from a clear yellow to a progressive brown, as found in the literature [16,18,24]. To remove Ag^+^, the solution was centrifuged at 4000 rpm for 20 min (Sigma 4-16 KS, Merck, Milan, Italy); the supernatant was discarded and replaced with the same volume of sterile ddH_2_O. This step was repeated three times. The purified nanoparticles were stored at room temperature in dark conditions and sonicated (30 kHz; SONICS Vibra cell, Sonics, Newtown, UK) immediately before the assays, three times for 15 s on and 45 s off. The nanoparticles derived from *C. longa* synthesis were referred to as ClAgNPs.

### 2.3. Biosynthesis of AgNPs Using a Cell-Free Extract of E. coli

#### 2.3.1. Preparation of Cell-Free Extract

The bacterial strain *E. coli* ATCC^®^ 25922 was used to synthesize silver nanoparticles. For the preparation of cell-free extract, the same conditions as those described by Kushawa et al. (2015) were used. Briefly, *E. coli* ATCC^®^ 25922 was cultured in Mueller-Hinton broth (Oxoid, Italy) and incubated aerobically at 37 °C until reaching the logarithmic phase of growth (assessed by spectrophotometric reading at 550 nm). The biomass was removed from the supernatant by centrifugation at 4000 rpm for 20 min (Sigma 4-16 KS); the liquid phase was filtered (0.22 µm syringe filter, Minisart^®^ Sartorius, Merck, Milan, Italy) to obtain the cell-free supernatant. To be sure about the absence of any residual bacteria in the supernatant, 10 µL was plated on Plate Count Agar (PCA, Condalab, Spain) and incubated aerobically at 37 °C for 24 h.

#### 2.3.2. Synthesis and Purification of Bacterial Silver Nanoparticles

Synthesis of AgNPs was carried out according to a method described previously [25,26]. A 1% AgNO_3_ (Carlo Erba, Italy) solution was added to the cell-free extract and incubated by shaking (200 rpm) for 24 h in a dark environment at 37 °C. The synthesized nanoparticles were centrifuged at 4000 rpm for 20 min (Sigma 4–16 KS) to remove the Ag^+^ and resuspended in ddH_2_O. This step was repeated three times. The nanoparticles derived from *E. coli* were referred to as EcAgNPs.

### 2.4. Characterization of AgNPs

The characterization of purified nanoparticles was carried out using two of the most used methods reported by different authors [5,18,24,25,27].

#### 2.4.1. Ultraviolet-Visible Spectroscopy (UV-vis)

Bio-reduction of Ag^+^ to Ag^0^ was monitored by measuring the absorption spectra (UV-vis) with a spectrophotometer (SpectraMax 340 PC, Molecular Devices, Munich, Germany) collecting spectra over a range of 310 to 770 nm (with an optical path length of 50 nm). Readings were recorded twice within 15 min.

#### 2.4.2. Transmission Electron Microscopy (TEM)

Size analysis of colloidal silver was characterized by EFTEM Leo 912ab (Zeiss, Milan, Italy) at a voltage of 100 kV. The samples were briefly sonicated (30 kHz, 15 s on and 45 s off), and immediately, a drop of the aqueous suspension of AgNPs was mounted on a carbon grid, which was placed on a filter paper to absorb the excess of solvent. The morphological analysis (particle diameter and size distribution) was calculated with Java image tool software (Image J2, v. 150).

### 2.5. Antimicrobial Properties

#### 2.5.1. Kirby–Bauer Disk Diffusion Assay

The antimicrobial activities of both ClAgNPs and EcAgNPs were tested against the 10 *P. aeruginosa* and 10 *S. pseudintermedius* strains and compared with antibiotics (carbenicillin and ampicillin, alone and in synergistic combination with silver nanoparticles) using the agar disk diffusion assay. Carbenicillin (Liofilchem, Roseto degli Abruzzi, Italy, 100 µg) was used against *P. aeruginosa* and ampicillin (Oxoid, 10 µg) against *S. pseudintermedius*. Briefly, after plating each strain on a sterile Mueller-Hinton agar plate (Oxoid, Italy), antibiotics and virgin disks were placed on the plates. In detail: two sterile disks were loaded with 20 μL of silver nanoparticles, two disks with antibiotics and two disks with antibiotic supplemented with 20 μL of silver nanoparticles, one disk was used as a negative control. All experiments were done in aseptic conditions in a laminar airflow cabinet, and two replicates were done for each strain. The plates were incubated aerobically at 37 °C for 24 h. Zones of inhibition for AgNPs, antibiotics, and antibiotics + nanoparticles were measured and expressed in millimetres (zone of inhibition ± SD).

#### 2.5.2. MIC

The minimum inhibitory concentration (MIC) was determined using the microdilution assay, according to the Clinical and Laboratory Standards Institute (CLSI) guidelines (CLSI, 2017). Briefly, all the strains were grown on Brain Heart Infusion agar (BHI, Scharlau, Milan, Italy,) and three or four colonies were suspended in fresh sterile saline solution to reach an initial concentration of 1.5 × 10^8^ CFU/mL. One hundred microliters of the 1:100 diluted cell suspensions was dispensed into each well of a 96-well microtiter plate. The 10 strains of *P. aeruginosa* were exposed to twofold dilution series of NPs alone and in synergistic combination (1:1 ratio) with carbenicillin. The same was done for *S. pseudintermedius* strains except for ampicillin instead of carbenicillin. To rule out that the antibacterial activity may be exerted by AgNO_3_ instead of AgNPs, MIC for AgNO_3_ was also performed (data not shown).

After incubation for 24 h at 37 °C, the MICs were determined as the lowest dilution of nanoparticles able to inhibit visible bacterial growth.

### 2.6. Antibiofilm Properties

#### 2.6.1. Screening for the Biofilm-Forming Ability

In a preliminary study (Appendix A), the biofilm-forming ability of the *P. aeruginosa* and *S. pseudintermedius* strains was studied using a microtiter plate assay (MtP) [28,29]. As a negative control, TSB + 1% glucose (TSBg) without bacteria was used. Each strain was tested in triplicate on three independent plates. Briefly, fresh overnight subcultures of the strains were 1:100 fold diluted in TSBg, and 200 μL were plated. After 24 h at 37 °C, the supernatant was gently removed and the biofilm was washed with sterile phosphate-buffered saline (PBS), fixed with methanol, and stained with 2% crystal violet. The absorbance (550 nm) of negative controls was used to set the optical density cut-off (ODc) as three standard deviations above the mean OD of the negative controls. Strains were classified as follows: non-adherent OD ≤ ODc; weakly adherent ODc < OD 97 ≤ 2 × ODc; moderately adherent ODc < OD ≤ 4 × ODc; strongly adherent OD > 4 × ODc. Only the biofilm-producer strains were used to study the anti-biofilm properties of silver nanoparticles.

#### 2.6.2. Interaction between AgNPs and Mature Biofilm

To determine the ability of both ClAgNPs and EcAgNPs to disrupt mature biofilm, an MtP assay using a volume of 190 μL of bacteria culture (after 1:100-fold final dilution), and 10 μL of AgNPs was carried out. The non-treated bacteria (NT) group was used to make a comparison with treated bacteria.

Using the above-described method [28], biofilm was grown at 37 °C for 24 h, then, 10 μL of nanoparticles was added, and the plates were further incubated for 24 h. The disruptive ability of the nanoparticles was determined by staining the biofilm biomass with crystal violet and measuring the absorbance at 550 nm (Labsystem Multiscan Plus, ThermoFisher Scientific, Milan, Italy). Each experiment was replicated three times per strain, and three independent plates were used.

### 2.7. Statistical Analysis

The statistical analysis, together with the graphical representation, were performed with GraphPad Prism version 7.00 for Windows (GraphPad Software^®^, San Diego, CA, USA, www.graphpad.com). To verify the normality (or non-normality) of the distributions, the following tests were performed: D’Agostino and Pearson omnibus normality test, Shapiro–Wilk normality test, and Kolmogorov–Smirnov normality test. For non-parametric distributions, the Mann–Whitney test was used to compare groups, otherwise the standard Student’s *t*-test. The results are presented as follows: means ± SD for disc diffusion assay; µM ± SD for MIC; OD ± SD for anti-biofilm activity. All data were tested for normality and analyzed using Student’s *t*-test and a *p*-value less than 0.05 was considered statistically significant.

## 3. Results

### 3.1. Synthesis of AgNPs

The change in colour (darkish-brown) of the silver solutions was used to monitor the bio-reduction of silver ions to silver nanoparticles (Figure 1). The intensity of brown increased over the time, as found in the literature. *C. longa* solution showed a more explicit darkish-brown appearance after the period of incubation (Figure 1A). On the contrary, the supernatant of *E. coli* turned immediately from yellow to yellow-white appearance without showing a brown aspect (Figure 1B). The confirmation of NPs synthesis and stability was monitored by UV-Vis spectroscopy, while the morphological analysis was performed with TEM.

#### 3.1.1. UV-Vis Spectroscopy

UV-Vis spectroscopy is considered one of the most important tools for the evaluation of NPs synthesis [24]. The absorption spectra of the AgNPs are shown in Figure 2. ClAgNPs showed a clear absorption peak at around 440 nm, confirming the production of colloidal silver nanoparticles. On the other hand, EcAgNPs showed a logarithmic-like curve which does not clearly show a specific absorption peak or suggest any NPs formation. To exclude instrumental bias, serial dilutions of EcAgNPs were loaded in the spectrophotometer with the same results in terms of peak’s absence.

#### 3.1.2. Size and Morphology Analysis of Silver Nanoparticles

Figure 3 shows the morphological analysis of silver nanospheres. It can be observed that NPs obtained with *Curcuma* showed a spherical symmetry and heterogeneous spatial dimensions ranging from 10 to 25 nm (11.1 ± 2.75 nm) (Figure 3A,C). The presence of narrow particles was in accordance with the UV-Vis results.

The analysis of *E. coli* supernatant showed nanoparticles enclosed in a matrix that connected each particle to the other (Figure 3B). These particles had a roughly spherical morphology and diameter ranging from 15 nm to 35 nm (27.28 ± 2.68 nm).

Overall, for both ClAgNPs and EcAgNPs, the results are in accordance with the literature [8,30], confirming that biological methods for the synthesis of AgNPs produce nanoparticles lacking homogeneity.

After the morphological analysis, the estimation of the molar silver nanospheres concentration was determined, as reported in the literature [14,31].

### 3.2. Antibacterial Activity

#### 3.2.1. Kirby-Bauer Disk Diffusion Assay

The potential synergistic effects of combining silver nanoparticles with antibiotics were evaluated with the disk diffusion assay (Figure 4). All the *P. aeruginosa* and staphylococcal strains were, respectively, resistant to carbenicillin and ampicillin. The antibacterial activity of both ClAgNPs and EcAgNPs increased when in combination with antibiotics in all the tested strains, showing significant differences (*p* < 0.05) in comparison with silver nanoparticles alone. For *P. aeruginosa* strains (Figure 4), ClAgNPs and EcAgNPs alone showed a mean inhibition halo of 9.5 and 14.4 mm, respectively, resulting in statistical differences when compared to carbenicillin + ClAgNPs (14 mm), and carbenicillin + EcAgNPs (17.45 mm). For *S. pseudintermedius* strains (Figure 4B), statistically differences were found between ClAgNPs and EcAgNPs alone (12.8 mm, and 16.4 mm, respectively), and in combination with ampicillin (19.61 mm, and 21.38 mm, respectively). The interpretation of Kirby-Bauer assay, using CLSI breakpoints, showed that all the staphylococcal strains were considered resistant to ampicillin even after the synergistic combination with NPs (resistance for ampicillin is <29 mm), while *P. aeruginosa* strains were resistant against carbenicillin alone, but when NPs were added, the strains were susceptible (resistance for carbenicillin is <17 mm).

The statistical differences between the two nanoparticles tested are shown in Figure 4, giving reasons to hypothesize a better antibacterial ability of EcAgNPs, but only in staphylococcal strains. This result is in contrast with the literature in which Gram-negative bacteria were reported as being more susceptible to silver nanoparticles than Gram-positive, depending on differences in their cell wall structure.

#### 3.2.2. MIC

MIC (Table 1 and Figure 5) of each tested strain was determined for both ClAgNPs and EcAgNPs alone and in synergistic combination with carbenicillin (for *P. aeruginosa*) and ampicillin (for *S. pseudintermedius*). As found in the Kirby-Bauer assay, both *P. aeruginosa* and *S. pseudintermedius* were resistant to carbenicillin (512 µg/mL) and ampicillin (56 ± 40.63 µg/mL). The results of NPs showed that MIC were different between Gram-negative and Gram-positive bacteria. When the antibiotics were adjuvated (1:1 ratio) with ClAgNPs and EcAgNPs, the MIC were significantly lower for both *P. aeruginosa* and *S. pseudintermedius*, resulting in susceptible profiles only for *Pseudomonas* strains (according to CLSI breakpoints). For both Gram-positive and Gram-negative bacteria, significant differences were found comparing the same antibiotic adjuvated with ClAgNPs and EcAgNPs and showing a lower MIC (better antimicrobial activity) for nanoparticles synthesized from *E. coli* supernatant.

Statistical differences were found, for *S. pseudintermedius,* between the combination ampicillin + ClAgNPs and the same antibiotic adjuvated with EcAgNPs (*p*-value *=* 0.002). Similarly, for *P. aeruginosa*, carbenicillin showed lower MIC when in synergistic combination with EcAgNPs instead of ClAgNPs (*p*-value *=* 0.0023).

### 3.3. Antibiofilm Properties

The formation of biofilm is a highly coordinated process during which bacteria adhere to each other and biotic/abiotic surfaces, resulting in the formation of a complex community enclosed in a self-produced extracellular matrix.

Following preliminary assays for the screening of biofilm-forming ability (Appendix A), 8 and 7 strains among, respectively, *P. aeruginosa* and *S. pseudintermedius*, were selected for their biofilm-forming ability.

These strains were grown for 24 h in microtiter plates and then exposed to ClAgNPs and EcAgNPs following a 1:20 dilution. The results (Figure 6) confirm the anti-biofilm ability of silver nanoparticles, but only for *P. aeruginosa* strains, the inhibition was statistically significant (*p* < 0.05) between the non-treated group and both nanoparticles. For staphylococcal strains, no differences were found between NT and both nanoparticles.

## 4. Discussion

In this study, the nanoparticles produced by *Curcuma* powder and *E. coli* showed heterogeneous dimensions, as observed in previous studies [16,18,25,26]. However, particle dimensions were different (11.107 ± 2.705 nm for *C. longa* and 27.282 ± 2.68 nm for *E. coli*) from those observed in the literature in which NPs ranged from 6.3–2.64 nm [16] to 50 nm [25]. One possible explanation could be the differences in terms of raw material used (e.g., quality of *Curcuma* powder) and the bacterial strain used as reported in the literature [32]. EcAgNPs showed a rough spherical morphology with a high-branched symmetry that could be responsible for the lack of the characteristic peak at 440 nm [33,34].

The disk diffusion assay demonstrated the ability of NPs (alone or in combination with antibiotics) to kill more easily antibiotic-resistant bacteria. In particular, statistical differences were found in S. *pseudintermedius* between nanoparticles from *E. coli* and *Curcuma* underlying the major effect of bacterial-derived nanoparticles. These results allow us to hypothesize that nanoparticles are broad-spectrum agents, as described in the literature [35]. It has to be noted that in the specific case of *S. pseudintermedius*, the strains that resulted were resistant to ampicillin also after the addition of nanospheres, whereas *P. aeruginosa* strains became susceptible to carbenicillin after NPs addition. The MIC values are in agreement with the literature in which NPs are reported as being more active against Gram-negative bacteria [18]. From the comparison of ClAgNPs and EcAgNPs between *S. pseudintermedius* and *P. aeruginosa*, only EcAgNPs were found to be statistically different in *S. pseudintermedius;* these results seem to be in contrast with MIC values, but could be explained by the presence of *E.coli* protein residues (used for the synthesis of EcAgNPs) present on the surface of NPs that could interfere with the staphylococcal growth [36]. It is well documented that the quorum-sensing pathways devoted to bacterial growth, communication and expression of virulence factors, are linked to the expression of species-specific peptides, experimentally peptides derived from Gram-negative bacteria can inhibit the growth or expression of specific proteins of Gram-positive [37]. One of the possible mechanisms by which silver nanoparticles exert their antimicrobial activity is the adhesion to the cell wall and subsequently breaking of this structure, leading the antibiotic to enter, and more easily continue to destroy the cell wall [7,19]. After the entrance, NPs can cause direct damages to proteins and DNA via reactive oxygen species (ROS) production [18].

One of the most studied bacterial virulence factors is the ability to produce biofilm through the quorum-sensing communication system. Biofilm development is related to environmental signals and communication systems that reflect on specific gene expression [38]. Nanoparticles from both *Curcuma* and bacterial synthesis can destroy the mature biofilm of *Pseudomonas*. The reason for this higher activity could be the adsorption of biomolecules on the surface of bacteria due to electrostatic attraction [14].

The synthesized AgNPs showed antibacterial activity alone and in combination with common antibiotics, suggesting a potential use as alternative therapies in case of Multi Drug Resistant-derived infections. However, their ability to destroy mature biofilm has to be properly studied in terms of localization and elaboration of other nanoparticles-based approaches.

## Figures and Tables

**Figure 1 vetsci-07-00012-f001:**
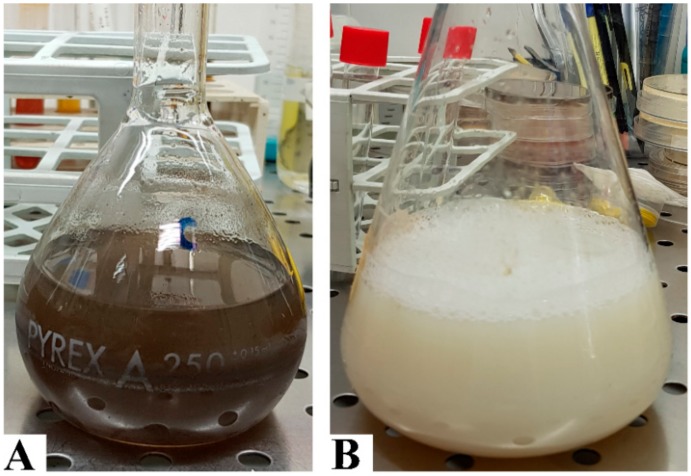
Biosynthesis of silver nanoparticles. Plant-mediated method (**A**) after dissolving *C. longa* powder in ddH_2_O, the solution was filtered to remove the majorities of vegetal debris; finally, silver nitrate was added and the change in colour was monitored. Bacterial supernatant method (**B**). After growing *E. coli* ATCC^®^ 25922 in Mueller-Hinton broth, the supernatant was collected, and silver nitrate was added to induce nanoparticle synthesis.

**Figure 2 vetsci-07-00012-f002:**
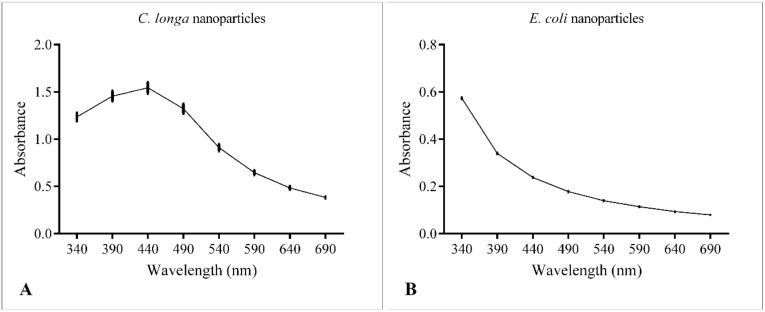
UV-Vis spectroscopy of ClAgNPs (**A**) and EcAgNPs (**B**).

**Figure 3 vetsci-07-00012-f003:**
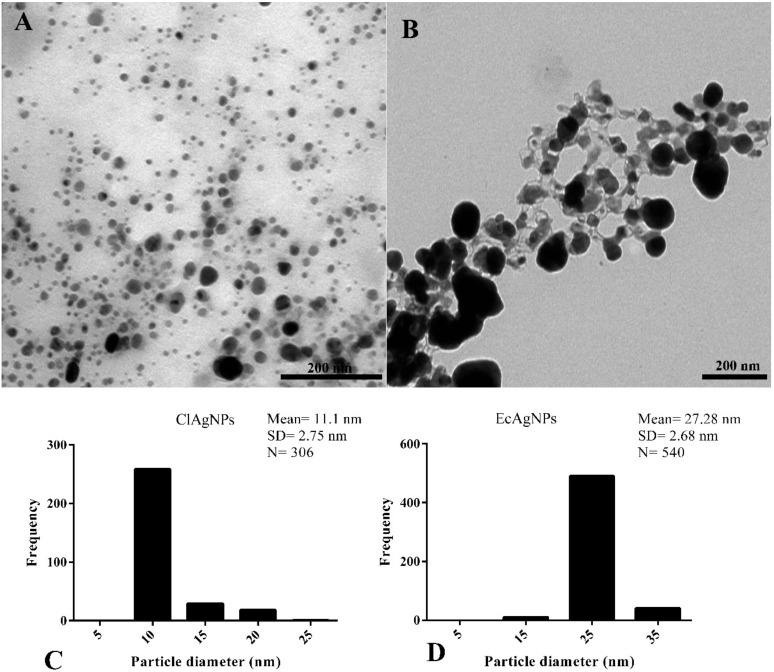
Morphological analysis by TEM. (**A**,**C**) NPs derived from *C. longa* showed roughly spherical morphology and mean diameter of 11.1 nm. (**B**,**D**) EcAgNPs showed a highly-branched symmetry and dimension of 27.28 nm in diameter.

**Figure 4 vetsci-07-00012-f004:**
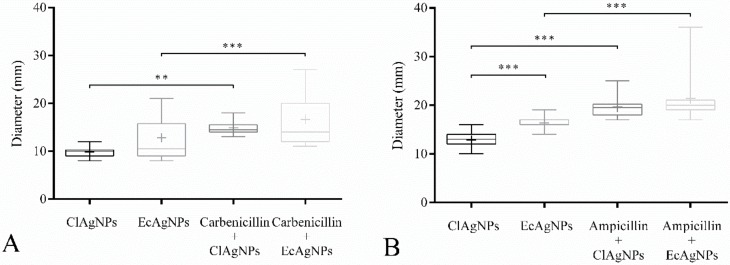
Disk Diffusion Assay. Effect of NPs used alone and in synergistic combination with (**A**) carbenicillin against *P. aeruginosa* and (**B**) ampicillin against *S. pseudintermedius*. Statistical analysis showed differences between inhibition halos of silver nanoparticles alone and those of silver in combination with carbenicillin and ampicillin. In *S. pseudintermedius* strains, EcAgNPs have a larger antibacterial effect compared to ClAgNPs. Results are presented as mean ± SD (* *p* value: 0.05–0.01; ** *p* value: <0.01; *** *p* value: <0.001).

**Figure 5 vetsci-07-00012-f005:**
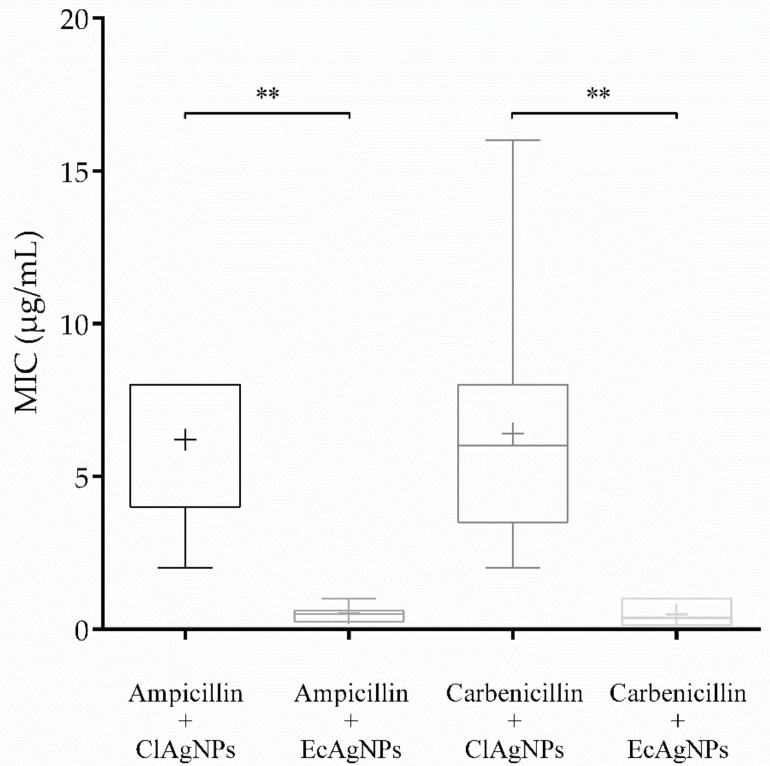
MIC of NPs in combination with ampicillin and carbenicillin (** *p* value: <0.01).

**Figure 6 vetsci-07-00012-f006:**
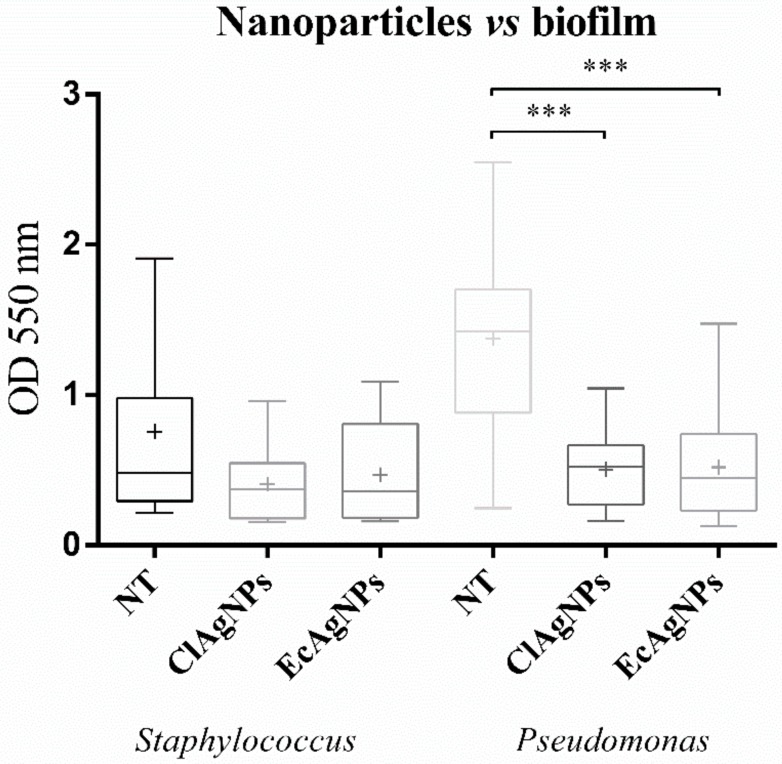
Anti-biofilm properties of NPs. Statistical differences were found only among *Pseudomonas* strains (*** *p*-value: <0.001).

**Table 1 vetsci-07-00012-t001:** Determination of MIC values for ClAgNPs and EcAgNPs alone and in combination with antibiotics. (carbenicillin 512–1 µg/mL for *P. aeruginosa*; ampicillin 256–0.5 µg/mL for *S. pseudintermedius*).

	*P. aeruginosa*	*S. pseudintermedius*
Antibiotic (µg/mL)	512 ± 0	56 ± 40.63
ClAgNPs (µM)	71.8 ± 0	143.7 ± 0
Antibiotic + ClAgNPs (µg/mL)	6.4 ± 4.2	6.2 ± 2.4
EcAgNPs (µM)	0.438 ± 0.18	3.75 ± 3.65
Antibiotic + EcAgNPs (µg/mL)	0.48 ± 0.37	0.5 ± 0.27

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
