# Peer review of "In Vitro Antibacterial Activity of Biological-Derived Silver Nanoparticles: Preliminary Data"

_vetsci, 2020, doi:10.3390/vetsci7010012_

Round 1
Reviewer 1 Report
Minor revision:
(i) Yield of the AgNP obtained.
(ii) Cyctotoxicity results of the synthesized AgNP.
Author Response
(i) Yield of the AgNP obtained.
Curcuma -> 8.36 nm - 22.5 nm
E. coli -> 11.55 nm - 30.04 nm
(ii) Cyctotoxicity results of the synthesized AgNP.
No cytotoxicity test have been performed.
Reviewer 2 Report
Meroni and co-workers report the antibacterial activity and biofilm eradication properties of biological-derived silver nanoparticles. Overall the manuscript is of reasonable quality: the experimental protocols and descriptions are well written which means interested readers could replicate this work. The data support the principle hypotheses made by the authors and the manuscript could be accepted following minor revision. My principle comments are as follows:
COMMENT 1. Why are the larger (ca. 27 nm) AgNPs more antimicrobial than their smaller counterparts (ca. 11 nm)?
If the differences can occur through surface charges of the particles then a more correct comparison would ClAgNPs vs. EcAgNPs of identical (or near identical) size; not 11.107±2.705 nm (ClAgNPs) vs. 27.282±2.68 nm (EcAgNPs)
Currently, it simply appears that the larger 27 nm EcAgNPs have more surface available Ag(0) from which Ag(I) ions can be liberated, in comparison to the smaller 11 nm ClAgNPs.
Some very recent reports have investigated the mechanism of action of biogenic AgNPs:
https://doi.org/10.1088/2053-1591/ab6636
Such studies should be cited in this manuscript and the authors should provide a more critical insight into their own results. Currently, it is clear WHAT the data show, but not WHY. Some logical hypotheses should be discussed.
COMMENT 2. For Figure 4, the KB disk diffusion assay (as well as all subsequent antibacterial assays, Figus 5-7), in my view the antibiotic + bacteria should included as a control, i.e. Carbenicillin for P. aeruginosa and ampicillin against S. pseudintermedius. Without this raw data it is impossible to compare the efficacy of the AgNPs. Although the strains are “antibiotic-resistant”, there is still a requirement to include such data for context and comparison. Of course, it is possible that the graphs may appear “odd” due to differences in scale between, for example, antibiotic + bacteria vs. AgNPs + bacteria, but such data should be included in Supporting Information at least.
Author Response
COMMENT 1. Why are the larger (ca. 27 nm) AgNPs more antimicrobial than their smaller counterparts (ca. 11 nm)?
If the differences can occur through surface charges of the particles then a more correct comparison would ClAgNPs vs. EcAgNPs of identical (or near identical) size; not 11.107±2.705 nm (ClAgNPs) vs. 27.282±2.68 nm (EcAgNPs)
Currently, it simply appears that the larger 27 nm EcAgNPs have more surface available Ag(0) from which Ag(I) ions can be liberated, in comparison to the smaller 11 nm ClAgNPs.
Some very recent reports have investigated the mechanism of action of biogenic AgNPs:
https://doi.org/10.1088/2053-1591/ab6636
Such studies should be cited in this manuscript and the authors should provide a more critical insight into their own results. Currently, it is clear WHAT the data show, but not WHY. Some logical hypotheses should be discussed.
We agree with you that the larger 27 nm EcAgNPs have more surface available Ag(0) from which Ag(I) ions can be liberated, in comparison to the smaller 11 nm ClAgNPs.
As for the paper you suggest (Estevez et al.,2019), when we submitted our work, this was still not published. But we will read it and use it as a reference for our discussion in the new version.
COMMENT 2. For Figure 4, the KB disk diffusion assay (as well as all subsequent antibacterial assays, Figus 5-7), in my view the antibiotic + bacteria should included as a control, i.e. Carbenicillin for P. aeruginosa and ampicillin against S. pseudintermedius. Without this raw data it is impossible to compare the efficacy of the AgNPs. Although the strains are “antibiotic-resistant”, there is still a requirement to include such data for context and comparison. Of course, it is possible that the graphs may appear “odd” due to differences in scale between, for example, antibiotic + bacteria vs. AgNPs + bacteria, but such data should be included in Supporting Information at least
In both cases the KB disk diffusion assay (with only the antibiotic) showed no measurable halo, making it impossibile a graphic rapresentation.
Reviewer 3 Report
Please see the attached review file.

Author Response
Introduction section----------
Line 11: The species names should be written in full (full genus) when first mentioned. This
should be done at the first mention in the text as well.
We'll revise the species names
Line 12: Please specify that Curcuma longa is turmeric.
We’ll specify either in abstract and in M&M sections.
Lines 20 – 21: Clarification needed in text – I presume these MIC values are in terms of micromolar total silver, assuming all of the silver is nanoparticles?
Generally, it is hard to quantitatively say how much silver is present unless elemental analysis of the cleaned-up nanoparticle solution is performed. Syntheses generally don’t go 100% to completion (as mentioned in the text, you wash the particles to remove excess Ag+), so it is difficult to estimate the amount of total Ag present based on the starting materials.
We agree with you, syntheses don’t allow a total conversion of Ag+ in AgNPs. For the quantification of nanoparticles produced (used to express MIC values) we used two different protocols found in the literature (Liu et al., 2007; Kalishwaralal et al., 2010) based on the extinction coefficient of silver and nanoparticles dimension (as reported in lines 236-237).
Line 24: “Antibiotic-resistance” shouldn’t have a hyphen – “antibiotic resistance” is fine as far as I know (I’m a native English speaker and haven’t seen it written the other way...Granted, maybe there’s a different convention in your part of the field?)
We’ll eliminate the hyphen.
Lines 28-29: Please put in some references about the emergence of nanomaterials in health. There are many, many, many review articles out there that likely would be suitable.
We’ll insert specific references.
Lines 43-44: Please put in some references to justify that sentence. Again, there’s a ton of review articles about nanomaterials synthesis that exist.
We’ll insert specific references.
Lines 44-46: Regarding the sentence “The physical approaches….” please add some general references. I suggest replacing the word “use” in line 46 with the word “include” because there are many physical methods beyond those mentioned – ball-milling comes to mind or vapor deposition.
We’ll insert specific references and replace the word “use” in line 45.
Line 51: Why does the release of proteins and secondary metabolites (in and of itself) make biological methods advantageous over chemical or physical methods? That does not make sense to me. I’d suggest getting rid of (a).
Proteins and secondary metabolites naturally released by metabolic pathways of living organisms act, at the same time, as reducing and stabilizing agents, avoiding the use of chemical reagents needed for these two steps of NPs production.
In the literature, some authors (e.g. Li et al., 2010; Sondi et al., 2004; Shamli et al., 2012) suggest the positive activity of these metabolites, in biological synthesis.
Line 52: “particles” should be changed to “particle”.
We’ll change it.
Line 58: Please put “the” in front of literature.
We’ll change it.
Line 59: I looked up reference 13 and it is only about gold nanoparticles (at least when I searched for the word “silver” in the article it didn’t show up). Perhaps you meant to put in a different reference?
Yes, reference 13 (Liu et al., 2007) refers to the method we’ve used to estimate the concentration of AgNPs. We made a mistake citing this reference in this part of the paper. We’ll delete it.
General: I strongly suggest that you justify why you are studying Pseudomonas aeruginosa and Staphylococcus pseudintermedius – what is their broader relevance to veterinary medicine? I think there needs to be more. P. aeruginosa is of course a problem in hospitals with MDR strains and biofilms on implants. I presume this is a problem in cattle operations or dairies where cows are fed antibiotics? S. pseudintermedius is a pathogen, but how common is it? I’m sure you can easily justify this, just add some references that give an idea about the scope of the problem (e.g. if most pyoderma is caused by these microbes?)
We decided to use Pseudomonas aeruginosa and Staphylococcus pseudintermedius first of all because they are relevant pathogen in either veterinary and human medicine; second because, in our microbiological practice, we frequently isolate P. aeruginosa from orthopedic and surgical infections and S. pseudintermedius from dermatological affection of dogs (mainly deep pyoderma). At least, these two pathogens were taken as model for Gram-negative and Gram-positive for the study of antibacterial activity of silver nanoparticles. This explanation had been added in the introduction.
Materials and methods section --------
The toxicological assays used are standard and straightforward.
I suggest for the materials characterization including dynamic light scattering (DLS) if at all possible. Even though you have a polydisperse mixture, it would give you a rough idea of whether you have aggregates in solution which can impact the toxicity and chemistry. TEM won’t do that. Also, TEM is not an ensemble measurement.
We wanted to perform DLS but unfortunately, we couldn't perform it.
Lines 76-83: Why was this characterization done, and why were those particular concentrations of antibiotics chosen?
I presume this is a standard method – just mention that with a reference and that should be fine.
Both the methods and the concentrations used, derive from our internal standard laboratory practice.
Line 91 – please put “the” in front of the word “literature”
We’ll correct it.
Re: statistical analysis – did you use ANOVA too? Please specify. Were all of your distributions normal (they don’t look like that to me from the box-and-whiskers plots)? If not, you have to do the non-parametric version of ANOVA and use the Mann-Whitney tests for your pairwise comparisons. Please specify what you did for non-normally distributed data (or if it was all normal).
No we didn’t use ANOVA test but we verified the normality (or non-normality) of distributions using the following tests: D’Agostino & Pearson omnibus normality test, Shapiro-Wilk normality test and KS normality test. For non-parametric distribution we used Mann-Whitney test, otherwise the standard T-test. This information will be added to “statistical analysis” section.
I was very confused by the multiple strains of these two microorganisms. For example, is the data you report for P. aeruginosa an average of the data from all strains?
Yes it is.
Lines 102-105: Did you do any testing to ensure that extra Ag+ was not being released during storage? This can readily happen, particularly when such samples are not refrigerated. One way to differentiate between nano-Ag and Ag+ is to use centrifugal ultrafilters, then to do elemental analysis on the filtrate and the retentate.
No we didn’t analyze the Ag+ release. But to provide an adequate stability of the nanoparticles, samples were refrigerated and all experiments were done within a week from the synthesis.
Results section --------------
The real focus of this work is the toxicology part, not the synthesis, because the synthesis is not new – you are just using the protocols. I suggest that some of your reporting of the synthesis and characterization go into the Supplementary Information.
We would like to keep this part in the material and methods section, but are open to move it to a supplementary section in case of necessity.
Line 198 – Please remove the mention of a “stronger synthesis” – it doesn’t make chemical sense. It isn’t necessarily straightforward to tell by eye whether you have a greater mass of nanoparticles given the changes in optical properties relative to size and aggregation.
We’ll delete it.
Figure 1: The “A” and “B” are really hard to see. Maybe enlarge and put in the top left-hand
corners?
We’ll format the figure.
Figure 2B: The result you show in figure 2B is troubling to me as from my recollection, a polydisperse solution of approximately spherical silver nanoparticles should simply result in a broadened peak around 410-440 nm. I suggest looking at the literature on how particle shape can impact the spectra.
We’ll add Helmlinger et al., 2016
Line 223-225: You already mentioned these details in the Materials and Methods – please delete.
We’ll delete it.
Line 229: Please capitalize the first word of the sentence.
We’ll correct it.
Line 230: Replace “seemed to have a spherical symmetry” with “had a roughly spherical morphology”.
We’ll correct it.
Lines 231-232: The sentence “Maybe this…” should be taken out. It is overly informal and I think scientifically not likely, not to mention that such a comment should go into the discussion. See comments on Figure 2B.
A short comment will be insert in the discussion and deleted from the results.
Lines 236-237: Molar concentration as in moles of nanospheres or moles of total silver? Do those papers concern highly polydisperse samples or monodisperse? If those methods were for nanoparticles with a narrow size distribution, you can’t say “determination” but rather “estimation”.
The concentrations refer to moles of nanoparticles. One of the two papers we have cited (Kalishwaralal et al., 2010) obtained polydisperse nanospheres.
Figure 3: Letters hard to see in A and B. Scale bars are impossible to read. Make a box with clear large text and a line with the same length as what is in the image to show this clearly. Why does B include that second image? It doesn’t make sense to me.
In the final figure, we will enlarge the scale bars and letters. In figure B the second image will be eliminated.
Lines 243-244: Replace “symmetry” with “morphology” and put “roughly” in front of “spherical”. Add a period.
We’ll correct it.
Figures 4 and 5: Very confusing!
You could make a single figure with two plots side by side.
Make one plot for P. aeruginosa that shows four categories next to each other: ClAgNPs, EcAgNPs, ClAgNP+carbnicillin, EcAgNPs+ carbenicillin.
Make a second plot for S. pseudintermedius that shows four categories next to each other:
ClAgNPs, EcAgNPs, ClAgNP+ampicillin, EcAgNPs+ ampicillin.
Please be sure to specify in the caption that you are presenting box-and-whisker plots, and please generate plots with the same formatting.
Also confusing: why again are you using carbenicillin and ampicillin? Was that due to the
antibiotic screening tests you mentioned in the materials and methods?
Thank you for the suggestions, in the final version we will insert a single new figure.
We used ampicillin and carbenicillin because the used strains resulted resistant during Kirby-Bauer assay and we aimed to study the synergistic effect of combining silver nanoparticles and antibiotic against which bacteria were initially classified as resistant. Moreover, both these molecules belong to the same pharmacological category and are commonly used in veterinary practice.
Line 278: Should be “Student’s t-test”.
We’ll correct it.
Line 287: “Sensible profiles”? I am not sure what these means – you are trying to state that with
NPs the microbes are more sensitive to the antibiotics?
Yes. P. aeruginosa is considered resistant to carbenicillin when MIC is >512 µg/mL and sensible for MIC <128 µg/mL. When carbenicillin is adjuvated with both ClAgNPs and EcAgNPs the MIC decreases to 72 and 0.48 µg/mL and the microorganism should be considered sensible. Unfortunately, this situation should not be stated for S. pseudintermedius. S. pseudintermedius is considered sensible to ampicillin when MIC <0.25 µg/mL, otherwise has to be classified as beta-lactamase producer. As stated in table 1, Staphylococcus has to be considered resistant also when ampicillin is adjuvated with nanoparticles. In line 287 we specify that sensibility it’s only for P. aeruginosa.
Table 1: I suggest replacing “Atb” with “antibiotic” – confusing to see yet another abbreviation show up.
“Atb” will be replaced with “antibiotic”
Figure 6: This figure is bizarre and not consistent with the others. I am confused by why there
are data points set horizontally next to each other. Use a standard box-and-whiskers plot?
The figure will be replaced with a standard box-plot.
Discussion section-----------
This, to my mind is the weakest section. Lines 321-329 should be removed – that is merely a summary of what was done and is not proper for a discussion. Lines 330-340 are irrelevant – it is a justification of the advantages of biological syntheses of nanoparticles. This should be removed.
Has suggested lines 321-340 will be eliminated.
The justifications as to the polydispersity of the syntheses (lines 341-346) are also irrelevant to the focus of the study, which ultimately is the toxicity of these nanoparticles and their combinations. It is well-known that biologically synthesized nanoparticles are (with a few exceptions) not as monodisperse as the chemically-produced nanoparticles.
We prefer to keep this part.
The most important aspect to address is why there is a synergistic effect of Ag NP with the antibiotics. Why do the Ag NPs potentiate the antibiotics (e.g. ROS are generated, breaking down cell membranes and allowing for Ag+ entry)? Is there anything in the literature about mixing Ag+ with antibiotics?
We tried to answer this question with a killing assay but unfortunately was done only with ClAgNPs and we preferred not to include the results in this paper. Silver nanospheres adhere to cell wall and break it (Li et al., 2009), leading the antibiotic to enter, and more easily continue to destroy the cell wall.
Secondarily, what is probably coating the EcAgNPs versus the ClAgNPs, and how would this make a difference in toxicity? There’s a lot of work from places like the Center for Environmental Nanotechnology (based out of Duke University) looking at how simply changing coatings can change the whole chemical behavior of a particle?
Yes, the coating surface determines specific and different interaction with microorganisms. Our hypothesis is that nanoparticles derived from E. coli could be partially coated with some peptides derived from E. coli itself that may result in negative interaction with S. pseudintermedius strains. Unfortunately, we didn’t found any answer to this question also in the literature.
Round 2
Reviewer 3 Report
Please see attached commentary file.
